# Long-Term Effectiveness of Physical Exercise-Based Swallowing Interventions for Older Adults with Dementia in a Day-Care Center

**DOI:** 10.3390/healthcare11091262

**Published:** 2023-04-28

**Authors:** Chia-Hui Chen, Chia-Yu Lin, Chiao-Ling Chen, Kuan-Ting Chen, Cho Lee, Ya-Hsin Yu, Chiao-Yu Shih

**Affiliations:** 1Department of Rehabilitation Medicine, Hualien Tzu Chi Medical Center, Hualien 970, Taiwan; 2Department of Psychiatric, Hualien Tzu Chi Medical Center, Hualien 970, Taiwan; 3Department of Physical Therapy, Tzu Chi University, Hualien 970, Taiwan

**Keywords:** dementia, swallowing safety, dysphagia, exercises-based interventions, day-care

## Abstract

Swallowing safety is one of the top health concerns of dementia. Coughing and choking (coughing/choking) are signs of impaired swallowing safety. This study aimed to investigate the effectiveness of regular physical exercise-based swallowing intervention for reducing coughing-choking at the dementia day-care center. This was a retrospective analysis with data from medical records, including age, the clinical dementia rating (CDR), and the frequencies of coughing/choking in ten days (10-day coughing/choking). Those who complied with the exercise programs were assigned to the exercise-based group (n = 22), and those who could not comply were assigned to the non-exercised-based group (n = 7). The non-exercised-based group showed more advanced age and higher CDR than the exercise-based group (*p* < 0.05). The 10-day coughing/choking showed significant decreases at the 5-month and 19-month in the exercise-based group and at the 5-month in the non-exercise-based group (*p* < 0.05). Our findings suggested that regular physical exercise-based swallowing intervention effectively alleviated coughing/choking problems of older adults with dementia and its effectiveness was long-lasting. For those who could not comply with exercise programs, noticeably with more advanced age and dementia, the effective swallowing intervention period was short-term.

## 1. Introduction

Dementia is a syndrome that leads to deterioration in cognitive functions beyond what might be expected from normal aging. Currently, dementia is the seventh leading cause of death and one of the major causes of disability and dependency among older adults worldwide [1]. The prevalence of dysphagia among older adults with dementia is 32~93%, as documented in previous studies [2,3]. Individuals with dysphagia are at a higher risk for developing airway penetrations, defined as entry of material into the laryngeal vestibule without passing below the level of the vocal cords, that could cause respiratory infections and aspiration pneumonia that lead to frailty and institutionalization, even to death [3,4]. Coughing and choking (coughing/choking) are warning signs of impaired swallowing safety. They can cause great distress to older adults living with dementia and their caregivers, along with a progressive decline in swallowing function [5].

The interventions for swallowing safety in dementia are usually cataloged into compensatory and therapeutic approaches [4,6,7]. Compensatory interventions include food/drink modifications, postural adjustments, and contextual measures, which are used to redirect food/drink flow to avoid the adverse effects of impaired swallowing physiology and prevent interruptions during mealtime to avoid distraction while eating. Compensatory measures are often necessary to minimize symptoms and consequences of dysphagia in the traditional view of dementia care. However, since these measures do not bring about direct alterations in the progressive decline of swallowing function, their impact is considered to be indirect. Swallowing-related exercises are a type of therapeutic approach that focuses on specific muscles involved in the swallowing process. By using targeted exercises, these methods aim to improve the ability to swallow safely and effectively, rather than just compensating for any deficiencies. These exercises have a direct impact on the swallowing process as they can bring about significant changes and enhance overall swallowing ability. To effectively manage dysphagia in neurodegenerative diseases such as dementia, a shift in perspective toward therapeutic approaches needs to happen because the current management approaches are primarily reactive or compensatory [6]. Regular physical exercise can offer older adults several benefits as they age, including better physical and cognitive function [8,9]. Hence, implementing regular physical exercise-based interventions for swallowing, which comprises direct benefits from swallowing-related exercises and indirect benefits from compensatory measures for swallowing function, can be a potentially effective therapeutic strategy for preserving or enhancing swallowing safety in older adults with dementia. However, limited evidence is available for regular physical exercise-based swallowing interventions used in dementia long-term care to prevent or reduce impaired swallowing safety, especially a lack of evidence on the long-term effectiveness for more than a year [10,11].

To implement therapeutic interventions for dementia, an interdisciplinary model is highly recommended because this model relies less on pharmacological treatment-based medical practice and more on integrated interventions from multi-disciplinary specialists and caregivers to deal with the complex range of cognitive, physical, socio-emotional, and self-care deteriorations associated with dementia [12,13]. The multi-disciplinary specialists often include geriatricians, neuropsychologists, nurse practitioners, physical/occupational/speech-language therapists, nutritionists, and social workers.

After a reformed long-term care (LTC) policy, called LTC 2.0, implemented in 2017, and a reformed dementia prevention and care policy, called Dementia Plan 2.0, established in June 2018, Taiwan’s long-term care system extended services to include hospital-affiliated dementia day-care units to provide integrated interventions for people living with dementia aged 50 years and more [14,15]. A physical exercise-based swallowing intervention for swallowing safety was initiated by an interdisciplinary team as a part of the daily treatment regimen in October 2018 at the dementia day-care center of Hualien Tzu Chi medical center in eastern Taiwan. This study aimed to investigate the long-term effectiveness of regular physical exercise-based swallowing interventions on swallowing safety among older adults with dementia at this day-care center.

## 2. Materials and Methods

### 2.1. Study Cases

This was a retrospective study with study cases obtained from medical records of the geriatric psychiatric day-care center (the dementia day-care center) at Hualien Tzu Chi medical center in eastern Taiwan. Information on the physical exercise-based swallowing intervention and data, including gender, age, and clinical data related to safe swallowing, were accessed on 34 individuals admitted to the dementia day-care center from October 2018 to May 2020. Five individuals whose data needed to be clarified were excluded. Twenty-nine individuals were included in the analysis.

The participation records of the physical exercise-based swallowing intervention were reviewed, showing that those who could comply with the instruction to perform physical exercises in the first month would extend their participation throughout 19 months, and those who could not cooperate to perform physical exercises in the first month would not comply with the physical exercise programs after. Of the 29 individuals, 22 who received the reactive swallowing interventions and complied with physical exercises were assigned to the exercise-based group. Seven who received the compensatory swallowing interventions and could not comply with the physical exercises were assigned to the non-exercise-based group.

The Research Ethics Committee at Tzu Chi General Hospital, Buddhist Tzu Chi Medical Foundation, approved this study (IRB110-137-B).

### 2.2. Physical Exercise-Based Swallowing Interventions

The physical exercise-based swallowing interventions consisting of swallowing-related exercises and compensatory measures were carried out before and during lunchtime every weekday from 11 October 2018 to 11 May 2020 (Figure 1). The program lasted 18 months and 21 days. This intervention was designed and implemented by an interdisciplinary team of geriatric psychiatrists, speech-language pathologists (SLP), physical therapists, nurse practitioners, and attending caregivers as a part of the daily treatment regimen of this dementia day-care center.

An SLP and the nurses in charge at the day-care center instructed the exercise-based group to engage in swallowing-related exercises for 20 min before lunchtime every weekday. The instructor-to-participant ratio was 1:3. The exercise programs, modified from Taiwan-SLPU swallowing exercises [16], included exercises of the head, neck, and shoulder, oral exercises of the lip and tongue, effortful swallow, effortful pitch glides, vocal function exercises, chin tuck against resistance, and breathing exercises. Table 1 presents a summary of the objectives and instructions for the swallowing-related exercises. For a more comprehensive description, please refer to Appendix A. The non-exercise-based group stayed in the same room and was free to do, see, or feel the activities of physical exercise programs.

A comprehensive compensatory intervention of six measures was checked on both groups during lunchtime every weekday: 1. keep the surrounding environment quiet and stable to minimize distraction while eating; 2. keep the oral environment intact to avoid loose dental braces or any obstacle, such as dry mouth; 3. keep sitting upright to avoid bolus pathway blockage; 4. use a small tablespoon (5 mL) for eating to avoid a bolus too large for each swallowing; 5. use a wide-open bowl but not a narrow-open cup for liquid food to avoid tipping the head back while drinking; 6. do not provide solid and liquid food simultaneously to avoid the confusing control of bolus flow. The mealtime compensatory measures were supervised by the nurses in charge, who prepared in advance and stayed with the participants during lunchtime to identify and fix any problems. The nurse-to-participant ratio was 1:3.

At home, the attending caregivers were given a health education leaflet on the six compensatory measures and were asked to check every action at mealtime.

### 2.3. Clinical Data Related to Safe Swallowing

The clinical dementia rating scale (CDR) [17] and the functional oral intake scale (FOIS) [18] were documented by geriatric psychiatrists. The CDR is a global 0–5 point rating scale for the severity of dementia (0 = absent; 0.5 = questionable; 1 = present, but mild; 2 = moderate; 3 = severe; 4 = profound; 5 = terminal). The FOIS was a 7-level scale: 1. no oral intake; 2. Tube-dependent with minimal/inconsistent oral intake; 3. tube supplements with consistent oral intake; 4. total oral intake of a single consistency; 5. total oral intake of multiple consistencies requiring special preparation; 6. total oral intake with no special preparation, but must avoid specific foods or liquid items; 7. total oral intake with no restrictions.

The frequency of coughing/choking in 10 days (10-day coughing/choking) was evaluated by nurse practitioners using the cumulative number at the day-care center. The occurrences of coughing/choking were tallied within a timeframe that included the duration of and 5 min following meals. Throughout this period, the participants were under the continuous supervision of attending nurses, with a nurse-to-participant ratio of 1:3.

As physical exercises of sustained phonation were expected to be beneficial for improving voluntary cough, maximum phonation time (MPT) [19] and coughing strength [20] were evaluated by SLP. Maximum phonation time (MPT) was a clinical measurement of the longest time one could phonate a vowel. Coughing strength was a clinical assessment of coughing sound and performance using a scale of 0~3: 0 (trace), 1(weak), 2 (fair), and 3 (good).

The 10-day coughing-choking was evaluated ten days before the beginning of the physical exercise-based interventions, and 5 months, 8 months, and 19 months after; the MPT and coughing strength were evaluated at the beginning of the physical exercise-based interventions and 5 months after; the CDR and FOIS were assessed at the beginning of the physical exercise-based interventions (Figure 1).

### 2.4. Statistical Analysis

The Shapiro–Wilk test was adopted to test the normality of continuous variables. Continuous variables were expressed as mean ± SD and the mean differences between groups were evaluated by independent *t*-test or Wilcoxon rank-sum test and within groups by paired *t*-test or Wilcoxon signed-rank test, depending on whether they followed a normal distribution or not. The chi-square test or Fisher’s exact test was used for the categorical variables to evaluate the group differences, expressed as frequency with proportion. The Wilcoxon signed-rank test was used to compare follow-up changes for each group. Statistically significant differences were defined as *p* < 0.05. All statistical analyses were performed using SPSS Statistics for Windows, version 17.0 (SPSS Inc., Chicago, IL, USA).

## 3. Results

### 3.1. Demographics and Clinical Characteristics in Two Groups

Table 2 shows the comparison of demographics and clinical characteristics at the beginning of the intervention. The exercise-based group included 22 cases with a mean age of 77.09 ± 6.84 years. The non-exercise-based group had seven cases with a mean age of 84.71 ± 3.90 years. The proportion of the exercise-based group was 75.8%. The non-exercise-based group was more advanced in age and CDR and less in FOIS, MPT, and coughing strength than the exercise-based group.

### 3.2. Short-Term Changes in 10-Day Coughing-Choking, MPT, and Coughing Strength

Table 3 showed the comparison of pre and post (5-month post) within and between groups regarding the mean of 10-day coughing/choking, MPT, and coughing strength. The exercise-based group had a significant decrease in 10-day coughing/choking and a significant increase in MPT at the 5-month post. Additionally, the increase in coughing strength of the exercise-based group was close to a significant level at the 5-month post. The non-exercise-based group significantly decreased in 10-day coughing/choking at the 5-month post.

### 3.3. Long-Term Changes in 10-Day Coughing-Choking

Table 4 shows the comparison between pre and post (5-month, 8-month, or 19-month) regarding the mean score of 10-day coughing-choking in two groups. The exercise-based group significantly decreased 10-day coughing-choking at the 5-month and 19-month follow-ups.

## 4. Discussion

The main findings of this study were that the regular physical exercise-based swallowing interventions at the dementia day-care center reduced the 10-day coughing/choking among those who complied with exercise programs at the 5-month and the 19-month follow-ups. Among those who could not comply with exercise programs, noticeably those with more advanced age and severe CDR, FOIS, MPT, and coughing strength, the compensatory intervention’s effectiveness for reducing the 10-day coughing/choking was marked at the 5-month post but declined after.

As nursing homes report a higher proportion of older adults with dysphagia than other settings, the evidence on interventions for dealing with swallowing safety in the demented elderly has mainly been based on the findings from these settings. According to a recent review, the most commonly implemented interventions to facilitate safe swallowing in nursing homes were “compensatory” in nature, such as modification of diet, ensuring an appropriate environment for swallowing, appropriate feeding assistance, and appropriate posture or maneuver for swallowing [10]. These compensatory swallowing measures often have a short-term benefit for safe swallowing through compensatory adjustments necessary to minimize the consequences of impaired swallowing function for those with difficulty swallowing. However, compensatory swallowing interventions do not address the underlying swallowing impairment. It has been suggested that only using “compensatory interventions” to manage swallowing safety may be problematic because they do not result in lasting changes in the underlying deterioration of swallowing functions, which is known to exist in those with ongoing neurodegenerative conditions [6,21]. Accordingly, for those who could not comply with exercise programs in this study, noticeably with more advanced age and worse dementia-related conditions, the effective compensatory swallowing intervention period for coughing/choking was marked in the first five months but subsided after. Based on the study’s findings, compensatory measures, such as modifying food/drink or posture during meals, can help alleviate coughing/choking right after implementing the swallowing intervention. However, these measures do not address the underlying deterioration of swallowing functions in the non-exercise group, which can result in the return of coughing/choking problems after a short period of time. Therefore, while compensatory measures are an important part of managing swallowing safety, they should be used in conjunction with efforts to directly address the underlying causes of the swallowing dysfunction, such as swallowing-related exercises.

Any type of physical activity is more beneficial than being physically inactive, and physical exercise confers greater benefits for physical functions [8,22]. In the study by Brach et al., older adults who participate in 20 to 30 min of moderate-intensity exercise on most days of the week have better physical function than those who are active throughout the day or inactive [22]. For older adults with dementia, more likely to be physically frail compared to age-matched controls [23], there was evidence that physical exercises improved their ability to perform activities of daily living [24,25]. Research suggests that exercise is advantageous for individuals who have dementia and, as a result, it is possible that swallowing-specific exercises could also be beneficial for those who have dysphagia and dementia. These conclusions were backed up by the results of this study, as well as previous research. In a review article by Langmore et al., exercise programs developed for improving swallowing functions were cataloged into swallow and non-swallow exercises [26]. The swallow exercises included effortful swallow, Mendelsohn maneuver, and the McNeill dysphagia therapy program. The non-swallow exercises included the Shaker exercise, lingual strengthening, expiratory muscle strength training, effortful pitch glides, vocal function exercises, chin tuck against resistance, and oral exercises. In a case series study by Balou et al., researchers demonstrated that swallowing physiology could be improved using a mixed swallow and non-swallow exercise protocol, including effortful swallows, tongue-hold swallows, supraglottic swallows, Mendelsohn maneuvers, Shaker exercises, and effortful pitch glides, in healthy adults with evidence of dysphagia [27]. To enhance the swallowing safety of older adults with dementia at the day-care center, we developed a mixed exercise program, modified from Taiwan-SLPU swallowing exercises, including relaxation exercises, oral exercises of the lip and tongue, effortful swallow, vocal function exercises, chin tuck against resistance, and breathing exercises. These physical exercises that target the muscles involved in swallowing can potentially maintain or enhance the functional reserve of swallowing. According to the findings of this research, older adults with dementia, whose ability to safely swallow often deteriorates with age, may benefit from regular 20-min physical exercise-based swallowing interventions, which combine swallowing-related physical exercises with compensatory swallowing measures, as it was found to increase their maximum phonation time (MPT) and coughing strength, and mitigate coughing/choking problems for up to 19 months. This highlights the potential of incorporating such interventions into the care plans for older adults with dementia to improve their swallowing function.

Dementia is a syndrome that causes cognitive and other related functions to become worse gradually, leading to confused behavior. It has been suggested that older adults with dementia may have difficulties participating in or performing physical exercise. A meta-analysis by Hong et al. reported that the average attendance rate of the exercise group among sedentary healthy older adults in 47 exercise RCTs was 86% [28]. Among older adults with dementia, the attendance rate of the exercise group was 83% from a study of moderate- to high-intensity exercise in community settings [29] and 75% from a survey of high-intensity exercise in nursing homes [30]. Similar to the findings from nursing homes, we demonstrated that the participation rate of physical exercises was 75% at the dementia day-care center. In the current study, we suggested that age and severity of dementia influenced physical exercise participation. Those who could not comply with exercise programs were of more advanced age and had worse symptoms of dementia. This was inconsistent with earlier findings [30,31]. Further investigation and discussion are required to elucidate the influence of age, the severity of dementia, or other factors on the participation in physical exercise among older adults with dementia and to find an appropriate solution to promote it at long-term care facilities.

Among multiple dementia LTC service models, day-care centers are designed primarily to meet the daily living and social needs of older adults with dementia during the day in a professional care environment to provide respite and support services for family caregivers [32]. Previous studies have suggested that good day-care services benefit older adults with dementia and family caregivers for overall well-being [33]. Under LTC 2.0 and Dementia Plan 2.0, hospital-affiliated dementia day-care centers in Taiwan usually extend the social-based services to include integrated interventions from multi-disciplinary medical specialists to deal with the complex range of health deteriorations associated with dementia. In a recent study from a hospital-affiliated day-care center in central Taiwan, researchers identified that the physical and cognitive functions of older adults with dementia were maintained or partially improved after a 6-month integrated intervention which included reminiscence, physical exercise, cognitive occupational, art, horticultural, and music therapies [34]. Swallowing safety was one of the top health concerns for persons with dementia and their caregivers, but a lack of articles addressed this issue at the day-care service. In our hospital-affiliated dementia day-care center in eastern Taiwan, a physical exercise-based intervention to enhance swallowing safety was developed by an SLP-led interdisciplinary team to be a part of the daily treatment regimen in the day-care center and resulted in a favorable outcome. For long-term care service, the hospital-affiliated day-care can become an integrated model to provide social-based respite and support services and medical-based interdisciplinary healthcare for older adults with dementia and their family caregivers.

There were several limitations of this study. First, some areas for improvement of a retrospective study existed in this study, including missing data in charts, loss of follow-up, and differences in baseline characteristics of the groups. Five cases were excluded due to largely missing data. Seven cases in the exercise-based group and one in the non-exercise-based group were lost at 19-month follow-up due to death or the COVID-19 pandemic. The group assignment in this study depended on compliance with the physical exercise programs in a day-care center. The results showed significant age, CDR, and FOIS differences between the exercise- and non-exercise-based groups. We recognized that the association of the physical exercise-based intervention and the 19-month mitigation in coughing-choking were not statistically significant after adjusting for confounding factors of age, CDR, and FOIS, meaning that it was impossible to determine whether the ease of coughing/choking was a result of physical exercise programs. Furthermore, the retrospective nature of this study may have some limitations with respect to participant recruitment and outcome assessment. The absence of a randomized control design and difficulties in recruiting non-exercising individuals within a clinical setting may have resulted in selection bias. Additionally, the subjective rating of cough strength/sound used in this study can be unreliable and biased, thereby affecting the study’s validity. The absence of FOIS and other outcome measures at various time points in this study further restricts the ability to assess the intervention’s overall long-term effects. Secondly, this study needed more sample sizes. A small sample size may decrease statistical power. This also limited further analysis by subgroups.

## 5. Conclusions

Impaired swallowing safety can be indicated by symptoms such as coughing and choking. An interdisciplinary intervention for swallowing, which combines compensatory swallowing measures and swallowing-related exercises, integrated into the daily treatment routine of a dementia day-care center, can effectively alleviate coughing and choking problems. The effects of this intervention may last for a significant duration. However, for older adults with severe dementia symptoms and advanced age who may struggle to adhere to exercise programs, compensatory swallowing measures may only provide short-term relief. It should be noted that our retrospective study could not establish a direct causal link between swallowing-related exercises and alleviating coughing/choking problems. Further randomized controlled studies are needed to determine the individual effectiveness of swallowing-related exercises in improving swallowing safety for older adults with dementia.

## Figures and Tables

**Figure 1 healthcare-11-01262-f001:**
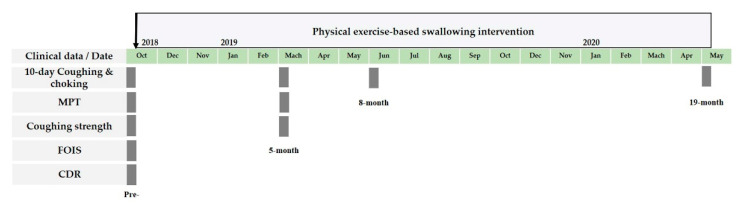
The chronological order of physical exercise-based swallowing intervention and clinical data (pre, 5-month, 8-month, and 19-month).

**Table 1 healthcare-11-01262-t001:** A summary of the objectives and instructions for the swallowing-related exercises.

Exercises	Objectives of the Exercises	Instructions for the Exercises
		Maintain a Straight Sitting Posture in the Chair Throughout All Exercises.
Exercises of the head, neck, and shoulder	To warm up and reduce tension in the muscles related to swallowing in the head, neck, and shoulder areas.	5–10 sets of neck forward stretching, side stretching, shoulder shrugging, and shoulder rolling.
Oral exercises of the lip and tongue	To maintain or enhance the functional reserve of the muscles required for the transition from chewing to swallowing.	5–10 sets of lip stretching, lip puckering, tongue protruding, and tongue side-to-side moving.
Effortful swallow	To maintain or enhance the functional reserve of the muscles used in swallowing.	5–10 sets of effortful swallowing.
Vocal function exercise	To maintain or enhance the functional reserve of the muscles involved in vocal fold regulation for swallowing safety.	5–10 sets of effortful pitch gliding, and sustained humming.
Chin tuck against resistance (CTAR)	To strengthen the suprahyoid muscles involved in opening the upper esophageal sphincter at the late stage of swallowing.	5 sets of sustained CTAR and repetitive CTAR.
Breathing exercises	To help improve respiratory function and support swallowing function.	5 sets of abdominal breathing.

**Table 2 healthcare-11-01262-t002:** Comparison of clinical characteristics between two groups.

	Total	Exercise-Based	Non-Exercise-Based	*p*-Value
N	29	22	7	
Proportion		75.8%	24.2%	
Age	78.93 ± 7.03	77.09 ± 6.84	84.71 ± 3.90	0.010 *
Gender	-	-	-	0.071
Male	11(37.9%)	6(27.3%)	5(71.4%)	
Female	18(62.1%)	16(72.7%)	2(28.6%)	
CDR	-	-	-	0.001 *
0.5	1(3.4%)	0(0.0%)	1(14.3%)	
1	15(51.7%)	15(68.2%)	0(0.0%)	
2	9(31.0%)	6(27.3%)	3(42.9%)	
3	4(13.8%)	1(4.5%)	3(42.9%)	
FOIS	6.31 ± 0.89	6.50 ± 0.86	5.71 ± 0.76	0.040 *
10-day coughing-choking	3.86 ± 5.38	3.55 ± 5.85	4.86 ± 3.72	0.583
MPT (s)	8.03 ± 5.17	9.27 ± 5.11	4.14 ± 3.18	0.019 *
Coughing strength	1.93 ± 0.81	2.09 ± 0.61	1.33 ± 1.21	0.041 *

CDR: dementia rating scale; FOIS: the functional oral intake scale; 10-day coughing-choking: the frequency of coughing-choking in 10 days; MPT: maximum phonation time; * indicates *p* < 0.05.

**Table 3 healthcare-11-01262-t003:** Comparison of pre and post (5-month post) within and between two groups regarding the mean of 10-day coughing/choking, MPT, and coughing strength in the exercise-based and non-exercise-based groups.

Item	Group	N	Pre	5M	Delta (5M-Pre)	Within Group *p*-Value	Between Group *p*-Value
10-day coughing-choking	Exercise-based	22	3.55 ± 5.85	1.41 ± 1.99	−2.14 ± 4.14	0.003 *	0.348
	Non-exercise-based	7	4.86 ± 3.72	2.71 ± 1.80	−2.14 ± 2.27	0.041 *	
MPT (s)	Exercise-based	22	9.27 ± 5.11	11.46 ± 5.77	2.19 ± 4.94	0.0497 *	0.713
	Non-exercise-based	6	4.67 ± 3.14	6.08 ± 3.65	1.42 ± 1.94	0.133	
Coughing strength	Exercise-based	22	2.09 ± 0.61	2.41 ± 0.65	0.32 ± 0.73	0.039 *	0.723
	Non-exercise-based	6	1.33 ± 1.21	1.50 ± 1.05	0.17 ± 0.41	0.317	

5M: 5-month; 10-day coughing/choking: the frequency of coughing/choking in 10 days; MPT: maximum phonation time; * indicates *p* < 0.05.

**Table 4 healthcare-11-01262-t004:** Comparison of pre and 5-month, 8-month, or 19-month post regarding the mean score of 10-day coughing/choking in the exercise-and non-exercise-based groups group.

Item	Group	N	Pre	5M	8M	19M	*p*-Value Pre vs. 5M	*p*-Value Pre vs. 8M	*p*-Value Pre vs. 19M
10-day coughing-choking	Exercise-based	22	3.55 ± 5.85	1.41 ± 1.99	1.86 ± 2.15	1.13 ± 1.67	0.003 *	0.091	0.040 *
	Non-exercise-based	7	4.86 ± 3.72	2.71 ± 1.80	2.86 ± 1.95	4.00 ± 3.41	0.041 *	0.246	0.892

5M: 5-month; 8M: 8-month; 19M: 19-month; * indicates *p* < 0.05.

## Data Availability

The data presented in this study are available on reasonable request from the corresponding author.

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
