# Peer review of "Long-Term Effectiveness of Physical Exercise-Based Swallowing Interventions for Older Adults with Dementia in a Day-Care Center"

_healthcare, 2023, doi:10.3390/healthcare11091262_

Round 1

Reviewer 1 Report

Page 2, study cases:

The characteristics of the experimental group that cooperated in participating in the exercise program and the control group that did not can be very different. This is a factor that can significantly affect the overall experimental results. Therefore, caution is needed in interpreting the results, and discussion of this in the review is necessary.

Page 3, Intervention:

It is not clear how the total exercise period for each participant is explained. Further explanation is needed for this

Page 4, Statistical analysis:

It is necessary to explain whether the normal distribution of the variables used in the statistical analysis was checked.

The difference between groups was tested with a paired t-test, but the follow-up change of each group was tested with Wilcoxon signed rank test. It is necessary to explain why a different statistic was used in a similar situation.

Page 5, line 168, significant increase in MPT at the 5-month follow-up: In the 'results' section, you should only talk about 'facts'. Strictly speaking, p=0.05(MPT) is difficult to consider as a statistically significant value.

In addition, it is the researcher's interpretation that the research result is close to 0.05 (coughing strength). It would be nice to talk about this in the Discussion section.

Since the number of experimental groups is nearly 4 times greater than that of the control group, even if the difference is the same, the experimental group can be statistically more advantageous. An explanation of this should be added in the discussion section.

There are cases where the p value is around 0.05 but not statistically significant. If the difference is judged to be clinically significant, it would be better to present the effect size together.

Page 7, line 259-260, MPT and and coughing strength were significantly increased at the 5-month follow-up in the exercise-based group: Statistically speaking, this is not true.

Author Response

Point 1: Page 2, study cases:

The characteristics of the experimental group that cooperated in participating in the exercise program and the control group that did not can be very different. This is a factor that can significantly affect the overall experimental results. Therefore, caution is needed in interpreting the results, and discussion of this in the review is necessary.

Response 1: Thank you for your comment and suggestion. The results of the study showed significant age, CDR, and FOIS differences between the exercise- and non-exercise-based groups. We recognized that the association of the physical exercise-base intervention and the 19-month mitigation in coughing-choking were not statistically significant after adjusting confounding factors of age, CDR, and FOIS, meaning that it was impossible to determine whether the ease of coughing-choking was a result of physical exercise programs. We discussed this limitation on Page 8, Lines 310-315 of the revised manuscript.

Point 2: Page 3, Intervention:

It is not clear how the total exercise period for each participant is explained. Further explanation is needed for this

Response 2: Thanks for your comment. The physical exercise-based swallowing interventions consisting of swallowing-related exercises and compensatory measures were carried out before and during lunchtime every weekday from October 11, 2018, to May 11, 2020. We addressed this on Page 3, Lines 104-106 of the revised manuscript. Besides, Table 1 of the revised manuscript offers a summary of the objectives and instructions for the swallowing-related exercises.

Point 3: Page 4, Statistical analysis:

  1. It is necessary to explain whether the normal distribution of the variables used in the statistical analysis was checked.

Response 3-1: Thanks for your suggestion. We had specified that we adopted the Shapiro-Wilk test to test the normality of each continuous covariate in 2.4. Statistical Analysis. Please refer to Page 5, Line 164 of the revised manuscript.

  1. The difference between groups was tested with a paired t-test, but the follow-up change of each group was tested with Wilcoxon signed rank test. It is necessary to explain why a different statistic was used in a similar situation.

Response 3-2: Thanks for your suggestion. We had specified that we adopted parametric tests or non-parametric tests depending on whether the covariates followed normal distribution in 2.4. Statistical Analysis. Please refer to Page 5 Lines 166-168 of the revised manuscript. FOIS, 10-day coughing/choking, and coughing strength did not follow a normal distribution in fact. Thus, we corrected the corresponding p-values in Table 2 and Table 3 of the revised manuscript.

  1. Page 5, line 168, significant increase in MPT at the 5-month follow-up: In the 'results' section, you should only talk about 'facts'. Strictly speaking, p=0.05(MPT) is difficult to consider as a statistically significant value.

Response 3-3: Thanks for your kind reminder. In fact, the p-value of testing whether there was a significant increase in MPT at the 5-month follow-up was 0.0497 < 0.05. We had corrected the p-value in Table 3 of the revised manuscript. It should be considered as a statistically significant value.

  1. In addition, it is the researcher's interpretation that the research result is close to 0.05 (coughing strength). It would be nice to talk about this in the Discussion section.

Response 3-4: Thanks for your kind reminder. We removed the wording from the footnote of Table 3 of the revised manuscript.

  1. Since the number of experimental groups is nearly 4 times greater than that of the control group, the experimental group can be statistically more advantageous even if the difference is the same. An explanation of this should be added in the discussion section.

Response 3-5: Thanks for your comment and suggestion. Because this is an observational study, it is difficult to recruit non-exercise-based cases in clinical settings. The great difference in sample size between the two groups was a limitation in this study. We had mentioned this limitation in the discussion section. Please refer to Page 8, Lines 315-318 of the revised manuscript.

  1. There are cases where the p value is around 0.05 but not statistically significant. If the difference is judged to be clinically significant, it would be better to present the effect size together.

Response 3-6: Thanks for your suggestion. We had adopted adequate statistical tests per your suggestion. Thus, we remained judging statistical significance via p-value.

  1. Page 7, line 259-260, MPT and and coughing strength were significantly increased at the 5-month follow-up in the exercise-based group: Statistically speaking, this is not true.

Response 3-7: Thanks for your comment. In fact, the p-value of testing whether there was a significant increase in MPT at the 5-month follow-up was 0.497 < 0.05. We had corrected the p-value in Table 3 of the revised manuscript. It should be considered a statistically significant value.

Reviewer 2 Report

This was a retrospective study of a group of participants with dementia who underwent a swallowing intervention program. The methods are not described in adequate detail and certain components of the methodology are not scientifically sound. As a result, I do not find the results to be robust enough to support the conclusions. I have provided specific feedback that I hope can help the authors in revisions and future research.

Title: I suggest you rename to "Long-term Effectiveness of Physical Exercise-based Swallowing Interventions on Swallowing Safety of older adults with dementia the Demented Elderly at a Hospital-affiliated Day-care Center"

Please note that the term "demented elderly" is not used in English clinical or academic language, I suggest changing this to "older adults living with dementia" in your manuscript. 

Introduction

- line 35: please briefly define what "penetrations" means in dysphagia, not all readers will be aware of dysphagia-specific terminology\

-line 43: please briefly define "bolus". This journal is for a generalist audience, dysphagia-specific terms will not be familiar to a lot of readers. You can just write "food/drink" to make it easier.

-line 46: I don't think it's accurate to say "Therapeutic interventions integrate intensive therapy programs into compensatory measures". While they are often used together, therapeutic interventions are their own category of dysphagia management strategies. They are the direct exercises compared to the indirect compensatory measures. It seems like you use the term "therapeutic program" based on your first definition in the rest of your manuscript. I highly recommend the authors make the distinction between indirect compensatory measures and direct swallowing-related exercises. If you want to use the term "therapeutic program" for all dysphagia management strategies, that's fine, but you need to separate indirect and direct measures first.

-line 48: reference [5] does not support the statement that contemporary management of dysphagia related to dementia has shifted. This paper points out that a perspective shift needs to happen because current management approaches are still reactive rather than therapeutic. 

-lines 50-51: please distinguish between the physical exercise researched in ref 8+9 and the swallowing specific exercises you are actually writing and studying. While they are all exercises, their targets are different, and you have to be careful in how you link them together. 

-reference 10 does not adequately support your research gap statement. A review like https://doi.org/10.1016/j.archger.2012.04.011 would be more appropriate as it focuses on dysphagia in people with dementia.

-line 67: "extended serviceS"

-line 68: "dementia day-care unitS"

-last sentence in introduction can be deleted

Materials and Methods

-there needs to be more details for each exercise: What does "relaxation" involve? How many repetitions of each exercise? How much resistance is provided during CTAR? How is the resistance provided? I suggest creating a table and including details for each exercise. I know exercises with this population is not usually organised, but your protocol for what you planned to do with them needs to be clear in the manuscript. Same with the meal-time compensatory measures: how were these monitored? Did someone sit with each participant? Were they constantly checking or every few minutes? Additionally, was the intervention carried out everyday? Every week day? 

-line 110: "WAS free to do"

-line 117: "...avoid tipping the head back while drinking"

- please cite the studies that designed the CDR and FOIS

- coughing-choking: so coughing and choking were grouped together? These can be very different and indicate different severity of the swallow dysfunction. How did you define choking? 

-10-day cough-choking: how do you ensure that the participant's every cough/choke is counted while at the centre? Is a nurse constantly with them? 

-subjective rating of cough strength/sound is extremely dubious and unreliable

- the lack of FOIS and other outcomes measures for all the follow-up periods is a major design flaw

-the methods are not clear on exactly how long the intervention program lasted. It seems like it was 19 months for those in the exercise group, but Figure 1 time labels call these "follow-up", which indicates they happened after the end of intervention. 

Results

-the 10-day cough-choke difference in the non-exercise group between pre and 5M is significant in Table 1, but not significant in Table 2 - check your analysis for errors.

-how do you determine improvement, maintenance, and deterioration? Is this based on the p-value?

-Table 4 is a bit hard to read. Consider adding more space between each time point.

-line 195-202 is unnecessary. You don't have to repeat exactly what your table already shows, you can try to describe the patterns if any.

-in general, I do not see much value in reporting the proportion of participants in terms of improvement/maintenance/decline, your groups are too small and imbalanced to begin with.

Discussion

line 242-243: same problem from intro, the evidence for exercise (ref 20+21) is not swallowing-specific exercise, you need link the two with your writing instead of equating them. E.g. There is evidence that exercise is beneficial for people with dementia. Therefore, swallowing-specific exercises could be beneficial for dysphagia in people with dementia. Research findings from this study and previous studies support this.

-Limitations - issues around the lack of all outcome measures at different time points and the use of subjective/unvalidated outcome measures need to be discussed

Author Response

Title

Point 1: I suggest you rename to "Long-term Effectiveness of Physical Exercise-based Swallowing Interventions on Swallowing Safety of older adults with dementia the Demented Elderly at a Hospital-affiliated Day-care Center"

Please note that the term "demented elderly" is not used in English clinical or academic language, I suggest changing this to "older adults living with dementia" in your manuscript.

Response 1: Thank you for your comments and suggestions. As suggested, we renamed the title: Long-term Effectiveness of Physical Exercise-based Swallowing Interventions on Swallowing Safety of older adults with dementia at a Hospital-affiliated Day-care Center. The term "demented elderly" was changed to "older adults with dementia" or "older adults living with dementia" throughout this manuscript.

Introduction

Point 1: - line 35: please briefly define what "penetrations" means in dysphagia, not all readers will be aware of dysphagia-specific terminology\

Response 1: Thank you for your comments and suggestions. The full term for "penetrations" is "airway penetrations". The definition of "airway penetrations" was added, and its source was cited as a reference [4]. The sentences were rewritten as follows: Individuals with dysphagia are at a higher risk for developing airway penetrations, defined as entry of material in the laryngeal vestibule without passing below the level of the vocal cords, that could cause respiratory infections and aspiration pneumonia that lead to frailty and institutionalization, even to death [3,4]. Please see Page 1, Lines 34-37 of the revised manuscript.

Point 2: -line 43: please briefly define "bolus". This journal is for a generalist audience, dysphagia-specific terms will not be familiar to a lot of readers. You can just write "food/drink" to make it easier.

Response 2: Thank you for your comments and suggestions. As suggested, we replaced "bolus" with "food/drink".

Point 3: -line 46: I don't think it's accurate to say "Therapeutic interventions integrate intensive therapy programs into compensatory measures". While they are often used together, therapeutic interventions are their own category of dysphagia management strategies. They are the direct exercises compared to the indirect compensatory measures. It seems like you use the term "therapeutic program" based on your first definition in the rest of your manuscript. I highly recommend the authors make the distinction between indirect compensatory measures and direct swallowing-related exercises. If you want to use the term "therapeutic program" for all dysphagia management strategies, that's fine, but you need to separate indirect and direct measures first.

Response 3: Thank you for your comments and suggestions. We rephrased sentences around "Therapeutic interventions integrate intensive therapy programs into compensatory measures" as follows: Compensatory measures are often necessary to minimize symptoms and consequences of dysphagia in the traditional view of dementia care. However, since these measures do not bring about direct alterations in the progressive decline of swallowing function, their impact is considered to be indirect. Swallowing-related exercises are a type of therapeutic approach that focuses on specific muscles involved in the swallowing process. By using targeted exercises, these methods aim to improve the ability to swallow safely and effectively, rather than just compensating for any deficiencies. These exercises have a direct impact on the swallowing process as they can bring about significant changes and enhance overall swallowing ability. Please see Page 5, Lines 48-56 of the revised manuscript.

Point 4: -line 48: reference [5] does not support the statement that contemporary management of dysphagia related to dementia has shifted. This paper points out that a perspective shift needs to happen because current management approaches are still reactive rather than therapeutic. 

Response 4: Point taken. We rephrased the sentence " Contemporary management of dysphagia related to dementia has shifted " as follows: To effectively manage dysphagia in neurodegenerative diseases such as dementia, a shift in perspective toward therapeutic approaches needs to happen because the current management approaches are primarily reactive or compensatory. Please see Page 5, Lines 56-59 of the revised manuscript.

Point 5: -lines 50-51: please distinguish between the physical exercise researched in ref 8+9 and the swallowing specific exercises you are actually writing and studying. While they are all exercises, their targets are different, and you have to be careful in how you link them together.

Response 5: Point taken. We rephrased the sentences with a linkage as follows: Regular physical exercise can offer older adults several benefits as they age, including better physical and cognitive function [8,9]. Hence, implementing regular physical exercise-based interventions for swallowing, which comprises direct benefits from swallowing-related exercises and indirect benefits from compensatory measures for swallowing function, can be a potentially effective therapeutic strategy for preserving or enhancing swallowing safety in older adults with dementia. Please see Page 5, Lines 59-64 of the revised manuscript.

Point 6: -reference 10 does not adequately support your research gap statement. A review like https://doi.org/10.1016/j.archger.2012.04.011 would be more appropriate as it focuses on dysphagia in people with dementia.

Response 6: Thank you for your suggestions. We replaced Reference 10 with "Alagiakrishnan, K.; Bhanji, R. A.; Kurian, M., Evaluation and management of oropharyngeal dysphagia in different types of dementia: a systematic review. Arch Gerontol Geriatr 2013, 56 (1), 1-9. doi:10.1016/j.archger.2012.04.011"

Point 7: -line 67: "extended serviceS"

Response 7: Thank you for correcting our grammatical errors. "extended service" was changed to "extended services".

Point 8: -line 68: "dementia day-care unitS"

Response 8: Thank you for correcting our grammatical errors. "dementia day-care unit" in line 68 was changed to "dementia day-care units".

Point 9: -last sentence in introduction can be deleted

Response 9: Thank you for your suggestions. The last sentence in the introduction was deleted.

Materials and Methods

Point 1: -there needs to be more details for each exercise: What does "relaxation" involve? How many repetitions of each exercise? How much resistance is provided during CTAR? How is the resistance provided? I suggest creating a table and including details for each exercise. I know exercises with this population is not usually organised, but your protocol for what you planned to do with them needs to be clear in the manuscript. Same with the meal-time compensatory measures: how were these monitored? Did someone sit with each participant? Were they constantly checking or every few minutes? Additionally, was the intervention carried out everyday? Every week day? 

Response 1: Thank you for reminding us of the need to provide more details about the exercise programs and compensatory measures. "relaxation" was an incorrect word to describe " Exercises of the head, neck, and shoulder", and we deleted "relaxation" in the revised manuscript. The repeat times of each exercise were added in the instructions portion of the newly created table (Table 1 in the revised manuscript). Chin tuck against resistance (CTAR): Hold a rubber ball under the chin with a hand and keep it in position during the exercise, chin tuck against the rubber ball as hard as possible. A table in the revised manuscript offered the purposes and instructions of each exercise (please refer to table 1 for a brief description and the table in supplement 1 for a detailed description in the revised manuscript). The mealtime compensatory measures will be supervised by the nurses in charge, who will prepare in advance and stay with the participants during the meal to identify and fix any problems. The nurse-to-participant ratio is 1:3. Please see 2.2. Physical Exercise-based swallowing Interventions in the revised manuscript.

Point 2: -line 110: "WAS free to do"

Response 2: Thank you for correcting our grammatical errors. "got free to do" was changed to "was free to do".

Point 3: -line 117: "...avoid tipping the head back while drinking"

Response 3: Thank you for your suggestion. "...avoid tipping the head back while drinking" was changed to "...avoid tipping the head back while drinking"

Point 4: - please cite the studies that designed the CDR and FOIS

Response 4: Thank you for your suggestion.

Reference 17 in the revised version was cited for the CDR.

Berg, L., Clinical Dementia Rating. Br J Psychiatry 1984, 145, 339.

Reference 18 in the revised version was cited for the FOIS.

Crary, M. A.; Mann, G. D.; Groher, M. E., Initial psychometric assessment of a functional oral intake scale for dysphagia in stroke patients. Arch Phys Med Rehabil 2005, 86 (8), 1516-20.

Point 5: - coughing-choking: so coughing and choking were grouped together? These can be very different and indicate different severity of the swallow dysfunction. How did you define choking? 

Response 6: Thank you for your comments. Coughing and choking can have distinct triggers, but in certain cases, they may co-occur. Therefore, using "coughing/choking" is more appropriate than "coughing-choking" to indicate the possibility of either or both of these actions happening. Hence, we will replace "coughing-choking" with "coughing/choking" throughout the revised manuscript.

When food/drink do not pass through the esophagus properly, they can trigger coughing or/and choking reflexes, depending on the severity of airway penetration and the individual's ability to clear their airway. In mild cases, a person may be able to clear the food or liquid by coughing. However, in more severe cases, the food or liquid may contact with or pass below the vocal folds and cause choking to prevent it from further deep into the airway. In some cases, coughing and choking reflexes may occur together. For example, if a person inhales a piece of food or a liquid, they may cough and experience a choking reflex simultaneously as their body tries to expel the foreign object and prevent it from entering the airway.

Point 6: -10-day cough-choking: how do you ensure that the participant's every cough/choke is counted while at the centre? Is a nurse constantly with them? 

Response 6: Thank you for your comments. The occurrences of coughing/choking were tallied within a timeframe that included the duration of and 5 minutes following meals. Throughout this period, the participants were under the continuous supervision of attending nurses, with a nurse-to-participant ratio of 1:3. This information was edited on Page 4, Lines 145-148 of the revised manuscript.

Point 7: -subjective rating of cough strength/sound is extremely dubious and unreliable

Response 7: Thank you for your comments. In the revised manuscript, we addressed limitation issues around subjective rating of cough strength/sound on Page 8, Line 315-323 of the revised manuscript.

Point 8: - the lack of FOIS and other outcomes measures for all the follow-up periods is a major design flaw

Response 8: Thank you for your comments. In the revised manuscript, we addressed limitation issues around the lack of FOIS and other outcome measures for all the post periods on Page 8, Line 315-323 of the revised manuscript.

Point 9: -the methods are not clear on exactly how long the intervention program lasted. It seems like it was 19 months for those in the exercise group, but Figure 1 time labels call these "follow-up", which indicates they happened after the end of intervention.

Response 9: The study's physical exercise-based swallowing interventions were carried out between October 11, 2018, and May 11, 2020. In research, "post" is commonly used to indicate the time point after an intervention has been administered in order to measure its effectiveness. The term "post" is more appropriate for this study than " follow-up". Hence, we replaced "5-month follow-up", "8-month follow-up", and "19-month follow-up" with "5-month post", "8-month post", and "19-month post" throughout the revised manuscript.

Results

Point 1: -the 10-day cough-choke difference in the non-exercise group between pre and 5M is significant in Table 1, but not significant in Table 2 - check your analysis for errors.

Response 1: Thanks for your kind reminder. We guessed that you meant the inconsistent p-values of 10-day cough/choke difference between pre and 5M in Table 2 & 3 for the non-exercise group. We checked the p-value of the 10-day cough/choke difference in the non-exercise group between pre and 5M in Table 2 & 3. We found there were some typos in Table 3. We corrected them. The p-values were consistent in Table 2 & 3 now. We really appreciate your help. Table 2 & 3 was changed to Table 3 & 4 in the revised manuscript.

Point 2:  

  1. -how do you determine improvement, maintenance, and deterioration? Is this based on the p-value?

Response 2-1: Thank you for your comments. Due to the limited significance of reporting the proportion of participants in terms of improvement, maintenance, or decline, we opted to eliminate section 3.4.

  1. -Table 4 is a bit hard to read. Consider adding more space between each time point.

Response 2-2: Thank you for your comments. Due to the limited significance of reporting the proportion of participants in terms of improvement, maintenance, or decline, we opted to eliminate section 3.4, including Table 4.

  1. -line 195-202 is unnecessary. You don't have to repeat exactly what your table already shows, you can try to describe the patterns if any.

Response 2-3: Thank you for your comments. Due to the limited significance of reporting the proportion of participants in terms of improvement, maintenance, or decline, we opted to eliminate section 3.4, including lines 195-202.

  1. -in general, I do not see much value in reporting the proportion of participants in terms of improvement/maintenance/decline, your groups are too small and imbalanced to begin with.

Response 2-4: Thank you for your comments. We agreed on your viewpoint that reporting the proportion of participants in terms of improvement, maintenance, or decline holds little value, as the groups are too small and unbalanced. Hence, we decided to eliminate section 3.4 which covers the long-term improvement and maintenance of the 10-day coughing-choking, including Table 4 and Figure 2.

Discussion

Point 1:  line 242-243: same problem from intro, the evidence for exercise (ref 20+21) is not swallowing-specific exercise, you need link the two with your writing instead of equating them. E.g. There is evidence that exercise is beneficial for people with dementia. Therefore, swallowing-specific exercises could be beneficial for dysphagia in people with dementia. Research findings from this study and previous studies support this.

Response 1: Thank you for your comment and suggestion. As suggested, " Research suggests that exercise is advantageous for individuals who have dementia, and as a result, it is possible that swallowing-specific exercises could also be beneficial for those who have dysphagia and dementia. These conclusions were backed up by the results of this study, as well as previous research." was edited to link the benefits of the exercises in ref 20+21 and swallowing-specific exercise in Page 7, Line 238-242 of the revised manuscript.

Point 2:  -Limitations - issues around the lack of all outcome measures at different time points and the use of subjective/unvalidated outcome measures need to be discussed

Response 2: Thank you for your comment and suggestion. The revised manuscript addressed the limitation issues related to "the absence of outcome measures at various time points" and "the use of subjective/unvalidated outcome measures". Please refer to Page 8, Line 315-323 of the revised manuscr

Reviewer 3 Report

Although the results of this study are limited by the sample size problem and the inability to exclude the effects of some confounding factors, their academic value is still worthy of evaluation. There are some inadequate descriptions of intervention methods and references, so please refer to the points below and revise them.

About intervention methods

Did the subjects attend this day-care center every day? Or just a few times a week?

How was the compensatory intervention performed at the day-care center? Verbal? in writing? Also, is it just once? Or was it done regularly or multiple times when staff noticed?

About references

Please add references to the sentence below.

P3, line 134 “Maximum phonation time (MPT) was a clinical measurement of the longest time one could phonate a vowel. Coughing strength was a clinical assessment of coughing sound and performance using a scale of 0~3: 0 (trace), 1(weak), 2 (fair), and 3 (good).”

Author Response

Point 1: Did the subjects attend this day-care center every day? Or just a few times a week?

Response 1: Thank you for your reminder. The subjects attended this day-care center every weekday. The physical exercise-based swallowing interventions consisting of swallowing-related exercises and compensatory measures were carried out before and during lunchtime every weekday from October 11, 2018, to May 11, 2020. We addressed this on Page 3, Lines 104-106 of the revised manuscript.

Point 2: How was the compensatory intervention performed at the day-care center? Verbal? in writing? Also, is it just once? Or was it done regularly or multiple times when staff noticed?

Response 2: Thank you for your reminder. The mealtime compensatory measures will be supervised by the nurses in charge, who will prepare in advance and stay with the participants during lunch to identify and fix any problems. The nurse-to-participant ratio is 1:3. Please see Page 3, Lines 127-130 of the revised manuscript

Point 3: About references

Please add references to the sentence below.

P3, line 134 “Maximum phonation time (MPT) was a clinical measurement of the longest time one could phonate a vowel. Coughing strength was a clinical assessment of coughing sound and performance using a scale of 0~3: 0 (trace), 1(weak), 2 (fair), and 3 (good).

Response 3: Thank you for your suggestion.

A reference regarding “Maximum phonation time (MPT)” was added to the reference19 :

Ko, E. J.; Chae, M.; Cho, S. R., Relationship Between Swallowing Function and Maximum Phonation Time in Patients With Parkinsonism. Ann Rehabil Med 2018, 42 (3), 425-432.

A reference regarding “Coughing strength” was added to the reference 20:

Ibrahim, A. S.; Aly, M. G.; Abdel-Rahman, K. A.; Mohamed, M. A.; Mehany, M. M.; Aziz, E. M., Semi-quantitative Cough Strength Score as a Predictor for Extubation Outcome in Traumatic Brain Injury: A Prospective Observational Study. Neurocritl care 2018, 29 (2), 273-279.

Round 2

Reviewer 1 Report

Thank you for revising the paper to reflect my comments.

Author Response

Point 1: Thank you for revising the paper to reflect my comments.

Response 1: Thank you for your comments and suggestions.

Reviewer 2 Report

Very impressive of the authors to have made these extensive amendments during the revision period, thank you for your efforts in addressing mine and the other reviewer's comments and questions. I think it was a good decision to trim the manuscript down and focus on the time points and outcomes that had sufficient data for reporting, now the manuscript is much more contained and cohesive. (Suggestion for future studies that include cough strength as an outcome: use a sound pressure level meter placed at a consistent distance from the lips, this is inexpensive and objective)

I'd like the authors to amend details about the length of the study and data collection time points before recommending accept:

- I suggest stating clearly in section 2.2, right after the start and finish dates of the intervention, that the program lasted X days/X months

- I don't think changing "follow up" to "post" really clarifies that all these time points occurred DURING the intervention, they both mean "after the end of the intervention" in my experience. I think just using 5M, 8M, is fine, and if 19M is immediately after the end of the intervention, then this would be "post". Or you can label 5M and 8M as interim data. Whichever label you decide on, please make these changes in figure 1 as well.

- with the amended labelling suggested above, you will need to edit the wording in the discussion: e.g. line 209-210 currently reads "...93.8% of them improved or maintained their coughing/choking problems at 19-month post." This will make readers think that 19 months after the intervention ended, the participants continued to improve or maintained their improvement gained from the intervention period, but 19 months is actually still within the intervention period/immediately after. These types of wording will need to be amended, please go through the discussion to check for all instances of this issue.

- also, where did the 93.8% come from? We don't have enough of the raw data reported to back this up, so I'd rather the authors stick with numbers that have already been reported in the results section

- discussion around the potential effects of the compensatory strategies for the non-exercise group's short term improvement is great

Author Response

Point 1: - I suggest stating clearly in section 2.2, right after the start and finish dates of the intervention, that the program lasted X days/Months

Response 1: Thank you for your suggestion. We added "The program lasted for a total of 18 months and 21 days." right after the start and finish dates of the intervention.

Point 2: - I don't think changing "follow up" to "post" really clarifies that all these time points occurred DURING the intervention, they both mean "after the end of the intervention" in my experience. I think just using 5M, 8M, is fine, and if 19M is immediately after the end of the intervention, then this would be "post". Or you can label 5M and 8M as interim data. Whichever label you decide on, please make these changes in figure 1 as well.

Response 2: Thank you for your comment and suggestion. As suggested, we made these changes (5M, 8M, 19M) in figure 1 Table 3, and Table 4.

Point 3: - with the amended labelling suggested above, you will need to edit the wording in the discussion: e.g. line 209-210 currently reads "...93.8% of them improved or maintained their coughing/choking problems at 19-month post." This will make readers think that 19 months after the intervention ended, the participants continued to improve or maintained their improvement gained from the intervention period, but 19 months is actually still within the intervention period/immediately after. These types of wording will need to be amended, please go through the discussion to check for all instances of this issue.

- also, where did the 93.8% come from? We don't have enough of the raw data reported to back this up, so I'd rather the authors stick with numbers that have already been reported in the results section

Response 3: Thank you for your comment and suggestion. "...93.8% of them improved or maintained their coughing/choking problems at 19-month post." came from the eliminated section 3.4. We edited out the wording related to the eliminated section 3.4 in the discussion and abstract in the latest manuscript.

Point 4: - discussion around the potential effects of the compensatory strategies for the non-exercise group's short term improvement is great

Response 4: Thank you for your suggestion. We added discussion in lines 230-237: Based on the study’s findings, compensatory measures, such as modifying food/drink or posture during meals, can help alleviate coughing/choking right after implementing the swallowing intervention. However, these measures do not address the underlying deterioration of swallowing functions in the non-exercise group, which can result in the return of coughing/choking problems after a short period of time. Therefore, while compensatory measures are an important part of managing swallowing safety, they should be used in conjunction with efforts to directly address the underlying causes of the swallowing dysfunction, such as swallowing-related exercises.
